# Unlocking New Avenues: Solid-State Synthesis of Molecularly Imprinted Polymers

**DOI:** 10.3390/ijms25105504

**Published:** 2024-05-18

**Authors:** Bogdan-Cezar Iacob, Andreea Elena Bodoki, Diogo Filipe Da Costa Carvalho, Antonio Augusto Serpa Paulino, Lucian Barbu-Tudoran, Ede Bodoki

**Affiliations:** 1Analytical Chemistry Department, Faculty of Pharmacy, “Iuliu Haţieganu” University of Medicine and Pharmacy, 4 Pasteur St., 400349 Cluj-Napoca, Romania; iacob.cezar@umfcluj.ro; 2Inorganic Chemistry Department, Faculty of Pharmacy, “Iuliu Haţieganu” University of Medicine and Pharmacy, 12 Ion Creangă St., 400010 Cluj-Napoca, Romania; abota@umfcluj.ro; 3Instituto Politécnico de Lisboa, Escola Superior de Tecnologia da Saúde de Lisboa, Av. D. João II, Lote 4.69.01, 1990-096 Lisboa, Portugal; diogoescola2013@outlook.pt (D.F.D.C.C.); antonio110495@gmail.com (A.A.S.P.); 4Electron Microscopy Center, Faculty of Biology and Geology, “Babes-Bolyai” University, 5-7 Clinicilor St., 400006 Cluj-Napoca, Romania; lucian.barbu@ubbcluj.ro

**Keywords:** mechanochemistry, liquid-assisted grinding, molecularly imprinted polymers, solid-state synthesis, molecular recognition

## Abstract

Molecularly imprinted polymers (MIPs) are established artificial molecular recognition platforms with tailored selectivity towards a target molecule, whose synthesis and functionality are highly influenced by the nature of the solvent employed in their synthesis. Steps towards the “greenification” of molecular imprinting technology (MIT) has already been initiated by the elaboration of green MIT principles; developing MIPs in a solvent-free environment may not only offer an eco-friendly alternative, but could also significantly influence the affinity and expected selectivity of the resulting binding sites. In the current study the first solvent-free mechanochemical synthesis of MIPs via liquid-assisted grinding (LAG) is reported. The successful synthesis of the imprinted polymer was functionally demonstrated by measuring its template rebinding capacity and the selectivity of the molecular recognition process in comparison with the ones obtained by the conventional, non-covalent molecular imprinting process in liquid media. The results demonstrated similar binding capacities towards the template molecule and superior chemoselectivity compared to the solution-based MIP synthesis method. The adoption of green chemistry principles with all their inherent advantages in the synthesis of MIPs may not only be able to alleviate the potential environmental and health concerns associated with their analytical (e.g., selective adsorbents) and biomedical (e.g., drug carriers or reservoirs) applications, but might also offer a conceptual change in molecular imprinting technology.

## 1. Introduction

Today, there is a special emphasis is on the “green chemistry” principles [1] as a step forward in polymer synthesis and functionalization. In recent decades, “green chemistry” has focused on finding the “ideal green solvent”, where the use of supercritical fluids, such as carbon dioxide [2,3], could at least partially address the environmental impact of large-scale organic solvent use and the generated chemical waste. However, due to several reasons, the alternative where solvents are eliminated entirely proved to be unattainable [4]. A promising direction in this respect is represented by mechanochemistry, an environmentally friendly chemical synthesis strategy that is carried out under solvent-free conditions or in the presence of a catalytic amount of solvent [5]. It was classified by the International Union of Pure and Applied Chemistry (IUPAC) as one of the top “10 chemical innovations that will change our world” [6]. Mechanochemistry refers to chemical reactions in a solid state that are triggered or influenced by mechanical forces [7]. In these reactions, mechanical energy is converted into chemical energy through the grinding, milling, or pressing of solid materials using various comminution devices. Examples of mechanochemical reactions include ball milling, sonication, and grinding of solid reactants to produce new compounds with unique properties. Such reactions have been reported for the synthesis of various inorganic [8] or organic molecules [9,10] or supramolecular structures [11,12], such as co-crystals, polymorphs, metal–organic frameworks, covalent-organic frameworks, and coordination polymers, with a plethora of scientific and industrial applications.

Mechanochemistry offers several advantages over traditional polymerization methods, including increased reaction rates, improved control over the molecular weight, and the ability to synthesize polymers under milder conditions [9]. The advantages of using mechanochemistry for constructive monomer-to-polymer synthesis over traditional polymerization, besides the method’s sustainability, are the increased reaction rates leading to significantly reduced polymerization times with higher yields; the lack of requisite expensive catalysts, starting materials, and appropriate solvents to dissolve all polymerization components; improved control over the molecular weight; mild reaction conditions; and last, but not least, the potential to unlock synthetic routes previously not accessible in solution [9].

A special type of functional polymer, whose synthesis and functionality are highly influenced by the nature of the solvent employed in its synthesis, is represented by molecularly imprinted polymers (MIPs). They are artificial receptors with tailored selectivity towards a target molecule, which are mainly employed in analytical applications as synthetic recognition elements [13,14,15], mimicking natural antigen–antibody interactions [16]. Lately, due to their inherent features, MIPs have also been explored as efficient synthetic catalyzers or as drug reservoirs for the controlled and even stimuli-responsive release of their payload in drug delivery systems [17,18].

The conventional synthesis of MIPs using the non-covalent approach is based on a free-radical-mediated chain-growth polymerization of vinyl monomers. The procedure is carried out in an appropriate solvent, often called a porogen, not only for dissolving all the components (i.e., template (target) molecule, functional and crosslinking monomers, and the initiator), but also to endow a porous structure to the polymeric matrix [19]. In a pre-polymerization step, template molecules and functional monomer(s) self-assemble into a complex, typically through non-covalent interactions. After polymerization, the template is removed, leaving behind binding pockets that are complementary in size, shape, and chemical functionalities to the template molecule. The resulting imprinted polymer will be able to selectively recognize the template molecule and its related structural analogues in subsequent rebinding experiments (Figure 1). To ensure the highest fidelity of the resulting binding pockets, the functional monomer–template complex must be preserved throughout the polymerization step; therefore, special attention must be paid to the polarity of the solvent used. Aprotic organic solvents with low polarities represent the appropriate medium for the stabilization of the hydrogen bonding, and in particular cases, electrostatic interactions can also be favored between the monomer and template molecules. However, these solvents are usually highly volatile and harmful to humans and ecosystems and represent the main type of residues from synthetic chemistry, both at the laboratory and industrial scales [20]. Protic solvents, like alcohols or water, competitively interfere with these interactions, and therefore, they tend to be avoided. All these problems could be alleviated by opting for mechanochemical synthesis, in which no solvent or very low catalytic volumes are required. Conventionally, solid-state synthesis may be achieved in simple comminution devices; however, more recently, mechanochemical reactivity has also been combined with other energy sources that are commonly used in wet chemistry, such as controlled heat exposure, light irradiation, sound or electrical impulses [21]. Therefore, as possible experimental approaches in the one-pot mechanochemical synthesis of the imprinted polymers either a (i) simple and cost-effective manual grinding of the reactants using a mortar and pestle (also known as “grindstone chemistry”) could be used, which, in turn, may be time- and labor-consuming and difficult to reproduce, or (ii) by using a ball mill, requiring more expensive equipment, and the commercially available grinding vials are currently not UV transparent. Nevertheless, as already demonstrated in [22], a vortex mixer is a proper alternative, combining the advantages of these two methods, as it is inexpensive and generate reproducible and simultaneous one-pot syntheses in small batches, which are highly desirable during the optimization phase.

Steps towards the “greenification” of molecular imprinting technology (MIT) has already been initiated by the elaboration of green MIT principles [23]; developing MIPs in a solvent-free environment may not only offer an eco-friendly alternative, but could also eliminate any potential interference with the imprinting process. Although a single recent report briefly compared the outcome of two green approaches, supercritical carbon dioxide technology and salt-assisted (SAG) mechanochemical grinding, for MIP synthesis and claimed successful molecular imprinting in both cases, the later demonstrated modest performance in terms of imprinting effect and binding capacity [24]. Furthermore, no comparison with conventional, solution-based molecular imprinting has irrefutably demonstrated the impact of the organic solvent as a critical variable in the experimental design since it simultaneously acts as a non-specific pore-forming agent and the dissolution media and also potentially interferes with the non-covalent interactions between key participants, namely the template and functional monomer(s), in the pre-polymerization phase. Additionally, any improvements in the imprinting factor without a selectivity assessment of the molecular rebinding process may give false perspectives about the applicability of MIPs for environmental and analytical purposes. Last, but not least, considerations regarding the environmental footprint and process operators (cost-, energy-, time-effectiveness) should also be taken into account in justifying the feasibility of lab-scale and/or large-scale transitions towards solventless molecular imprinting.

In this regard, solvent-free mechanochemical synthesis of MIPs was carried out using a hydrophilic model molecule (atenolol (ATNL)), which is from a class of templates that is still raising challenges in conventional non-covalent molecular imprinting. To offer first-hands projections on their analytical capabilities, the imprinting efficiency and selectivity have been evaluated using rebinding studies, in relation to other cognate drugs and to the corresponding MIP prepared using the conventional imprinting protocol.

## 2. Results and Discussion

### 2.1. Synthesis and Characterization of Polymers

In this study, ATNL was selected as the model template molecule for the solid-state synthesis of molecularly imprinted polymers using the non-covalent approach, based on a previously reported with some modifications [13]. Considering ATNL’s basic nature that promotes interactions with acids, two acidic functional monomers were employed, namely methacrylic acid (MAA) and 2-(trifluoromethyl)acrylic acid (2-TFMAA), which can form hydrogen bonds and weak electrostatic interactions with the 2-hydroxylethylamine group of ATNL [25,26]. MAA is probably the most commonly used monomer in imprinting technology, while 2-TFMAA is a stronger acid because of its inductive electron-withdrawing trifluoromethyl group (–CF_3_) that is capable of forming more stable intermolecular complexes with the template. Although it not an ideal experimental design for a comparative study, due to the solubility issues of the cross-linker, methylene bis-acrylamide (MBA), in the porogenic solvent (acetonitrile, ACN), in the wet MIP synthesis, MBA was replaced with ethylene glycol dimethacrylate (EGDMA). On the other hand, employing the liquid EGDMA in the mechanochemical synthesis would lead to a paste-like consistency, which hinders the energy transfer during the grinding process and thus renders the process ineffective [27]. Nevertheless, even though the cross-linker represents the major component of the resulting MIP scaffold, the selective molecular recognition of the template is expected to be mostly related to the employed functional monomer. In the first attempt at mechanochemical MIP synthesis, the same molar ratio of template/monomer/cross-linker was used as in the wet molecular imprinting approach. As all the key components of the polymerization mixture (namely the template, functional monomer, cross-linker, and radical initiator) are water soluble, during the optimization of the mechanochemical conditions, the chain growth and formation of the polymer were visually monitored through the emerging fraction of the insoluble reaction product. The effect of small volumes of solvent, the use of co-milling agents (i.e., alumina), the milling time, and the benefits of photoinitiation were investigated as primary variables affecting the yield of radical polymerization products.

Radical polymerization using mechanochemistry involves three major sequential phases. During the initiation step, the mechanical energy generated by milling or grinding in the comminution device (e.g., using a mortar and pestle, the collision of metallic, ceramic, or glass balls) causes the formation of free radicals, which act as initiators for the polymerization reaction. Additionally, in all our experimental setups, ~10% w/w APS was also added as a co-milling agent. APS, which is also used in conventional MIP synthesis as a radical initiator, under the mechanical effect should generate consistent amounts of SO_4_^•−^ through homolytic cleavage. The propagation of free radicals is sustained by their reaction with the monomers, leading to a chain growth process, which is eventually terminated by the reactions between the radicals, resulting in the formation of non-reactive species and the termination of the polymerization reaction [9]. As the propagation of radicals under neat grinding conditions may be more tedious, the use of very small volumes of solvent (ratio of liquid volume to reactant weight, η, is ~0–2 µL/mg), also called liquid-assisted grinding (LAG), can significantly accelerate the reaction rate or may even enable reactions that do not proceed at all in its absence. Although the parameters (e.g., solvent polarity) controlling or predicting the outcome or efficiency of LAG are far from being fully understood, this technique is often used by mechanochemists to optimize solid-state reactions [28]. Using LAG in the synthesis process led to a significant increase in polymer yield under all conditions (with or without alumina, UV exposure, etc.) in comparison with neat grinding. Oxides, such silicon dioxide or alumina [29,30], are often used as mechanoactivated catalytic surfaces in solid-phase synthesis. Even though alumina could additionally enhance the radical initiation and propagation through ·OH formation [30], its admix in the current mechanochemical setup turned out to not be critical, as polymer formation was observed regardless of its use (MIP synthesis, Appendix A, ESI). The milling time obviously had a critical influence on the progress of the polymerization, which was dependent on the energy of the mechanical input and the complexity (number and molar ratio of components) of the milling mixture, with the type and yields of the ongoing chemical reactions showing strong variations. The initial experiments and former reports on the mechanochemical polymerization of MAA [31] demonstrated acceptable yields (5–40%) within tens of minutes to up to a few hours; thus, in the UV-assisted mechanochemical synthesis of ATNL@MIP, 2 h was selected as standard milling time using two 4 mm stainless steel balls in glass vials vortexed at 3000 rpm. Interestingly, the yields of the polymerization product using the mechanochemical approach were higher for the imprinted polymers compared to the non-imprinted ones.

The imprinted polymers prepared using both the mechanochemical approach and the conventional solution-based polymerization approach were characterized by both scanning electron microscopy (SEM) (Appendix A, ESI) and FT-IR spectroscopy (Appendix A, ESI). The micrographs of both the MIP and NIP particles prepared via vortex milling (Appendix A–H, ESI) exhibited morphological similarities, presenting structures with irregular shapes and a broad size distribution. The size of the majority of these particles were in the micrometer range, being larger and with a smoother surface than the polymers prepared in an organic solvent (Appendix A–N, ESI). No obvious difference in terms of morphology could be observed between the imprinted polymers and their corresponding non-imprinted polymers when using different functional monomers.

Because precipitation polymerization was used in the conventional synthesis approach, spherically shaped polymer particles were obtained in all cases. Monodisperse porous microparticles with a mean diameter of ~1 μm resulted when MAA was used as the functional monomer, which was slightly bigger than the corresponding MIP prepared with 2-TFMAA which, in turn, showed a more aggregated structure. The NIPs obtained using the organic solvents were homogeneous spheres with particle diameters smaller than 50 nm.

ATR-FT-IR spectroscopic measurements performed on the washed and dried reaction products obtained from the mechanochemical synthesis also confirmed the formation of acrylate-based polymers. The characteristic vibrational band assignment of the key components of the reaction mixture, namely the two functional monomers, MAA and 2-TFMAA, as well as the employed cross-linker, MBA, was performed based on literature data [32]. As expected, the spectral profiles of the MIPs and NIPs were identical (Appendix A, ESI), as both types of polymers were subjected to the same (template) washing procedure. The obtained imprinted and non-imprinted polymers exhibited IR spectra closely resembling those reported in the literature for similar acrylate-based polymers [31,33,34]. As major component, most of the strong characteristic vibrational bands of the pure cross-linker (MBA) were also found in the spectra of the resulting polymeric products. These bands were slightly widened due the lengthening of the polymeric chain and their involvement in additional hydrogen bonding, such as the C=O stretching mode (Amide I band, 1650 cm^−1^), N–H deformation (amide II band, 1535 cm^−1^), N–H stretching (3305 cm^−1^), and C–N stretching mode (amide III band, 1301 cm^−1^) [32,35]. However, the full suppression of the vinylic group’s stretching mode (~1620–1630 cm^−1^) present in the spectra of the pure cross-linker (1621 cm^−1^ for MBA) and functional monomer(s) (1630 cm^−1^ for MAA; 1632 cm^−1^ for 2-TFMAA) is a strong indication of the polymeric network’s formation [32,36]. Particularities in the IR spectral profiles of the imprinted and non-imprinted M1 and M2 polymers were related to the type of functional monomer employed (MAA and 2-TFMAA) and were identified at ~1180 cm^−1^ due to the intense stretching mode of the highly polar C-F bond (Appendix A, ESI) [37]. The FTIR spectral analysis showed no significant variances in band intensities or their ratios when comparing MIPs with the corresponding NIPs, suggesting comparable polymer compositions and levels of crosslinking in both polymer types.

To evaluate the environmental impact of chemical synthesis processes in a simple and straightforward manner, the American Chemical Society Green Chemistry Institute developed an assessment tool based on the 12 principles of green chemistry [1], called DOZN 2.0, which was made available by Merck^®^ as a web-based scoring matrix. When comparing wet chemical synthesis with solid state, mechanochemically activated synthetic approaches, DOZN 2.0 considers factors such as energy consumption, waste generation, and hazardous materials. Such tools can quantitatively evaluate and compare the environmental performance of different synthesis methods, aiding in the development of more sustainable and environmentally friendly chemical processes [38]. The obvious difference in the obtained aggregate scores clearly indicates that the mechanochemical batch synthesis process was greener in comparison with the solution-based approach (Table 1, Appendix A, ESI). Moreover, the environmental footprint of the wet polymer synthesis technique significantly increased with the production batch size due to the need for larger quantities of the organic solvent.

The difference in scores (1 vs. 12) was mainly due to the difference in the resource use, which is driven by resource efficiency and waste reduction, and in the reduction in human and environmental hazards, which address the potential impacts to human health and safety and the environment. The employed nonpolar solvents in conventional imprinting synthesis are usually highly volatile and harmful to humans and ecosystems and represent the main type of residues from synthetic chemistry, both at the laboratory and industrial scales [20]. The score for the increased energy efficiency principle was 0 in all cases. As the synthesis time and output mass compared to the raw materials, including the need for a solvent, is detrimental for the solution-based methods, and the corresponding aggregate scores were higher in all cases for the wet MIP synthesis. Nevertheless, as the amount of solvent used in LAG reactions is much too small to become an environmental concern, LAG is considered a solvent-free technology. No follow-up processing (template and residual reagent elution) was considered, as this should be identical for the products of both types of processes. As expected, no significant differences were observed between the aggregate scores of the two imprinted (M1 and M2) and two non-imprinted polymers (NIP 1 and NIP2) within a given type of synthesis process (wet or solid-state), as the reaction conditions and overall yields calculated based on the output mass of the water-insoluble polymers were quite similar.

### 2.2. Binding Experiments

The rebinding performance of the synthesized polymers towards the template molecule was assessed using equilibrium binding experiments with 100 μg/mL ATNL in water. Considering the aqueous nature of biological and environmental samples commonly used for atenolol determination, the rebinding studies were conducted in a water-based environment. Based on our prior research [25,26] and due to the weakly basic nature of the studied beta-adrenergic receptor antagonists (conferred by the secondary amine in the alkyl side chain) within the neutral to slightly acidic range of ultrapure water, all the investigated analytes, including the template itself (atenolol, pKa = 9.43), were present in their protonated form during the rebinding process. This protonated form may engage in electrostatic interactions with the negatively charged polymer matrix. Additionally, other weaker interactions such as hydrogen bonding, van der Waals forces, etc., were contributing and most probably modulating the overall rebinding process within the MIP.

The adsorption capacities of the MIPs and NIPs were determined by calculating the amount of b-blocker mass adsorbed per gram of polymer, despite the distinct morphologies observed in MIPs and NIPs. These morphological differences are intrinsic to the imprinting process and must be taken into account when making comparisons between MIPs and NIPs.

The adsorption capacities of both MIPs prepared by mechanochemistry were much higher compared to the corresponding non-imprinted polymers (Appendix A, ESI). NIP_M2 showed a more than a three-fold increase in the amount of bound template compared to NIP_M1, suggesting a higher non-specific adsorption, probably due to the stronger acidic character of 2-TFMAA. It is worth highlighting that MIP_M1 and MIP_M2 exhibited similar or better adsorption capacities as MIPs prepared using the conventional solution-based protocol. However, the corresponding NIP_Ss yielded a very high non-specific adsorption, which led to rather poor imprinted factors (IFs) for the MIP_Ss, most probably due to the significantly lower particle sizes and higher surface-to-volume ratios of the non-imprinted counterparts. MIP_M1 showed the highest IF, three times higher than that of MIP_M2 and more than eight times greater compared to that of the same polymer prepared in the organic solvent.

The use of supercritical fluids as eco-friendly dispersion media in the molecular imprinting of small hydrophilic templates (i.e., amino acid) may also lead to noteworthy imprinting efficiencies in comparison with the conventional, solution-based non-covalent imprinting approach [24]. This alternative molecular imprinting approach may have a series of unforeseen advantages once the mechanistic insights and the impact of the employed solvent are unveiled. Nevertheless, in order to gain a wider acceptance as a feasible molecular imprinting tool, some considerable limitations still need to be overcome, such as the limited accessibility to more sophisticated instrumentation, the limitations of polymerization temperatures, the solubility issues of key ingredients in the supercritical fluid, etc.

The selectivity of the molecular recognition process, as one of key features of the synthesized MIPs, was investigated using different structural β-blocker analogues of the template, which share a common alkyl side chain with different aromatic rings and additional functionalities attached to it, such as propranolol (PRNL), alprenolol (ALPRNL), and carvedilol (CVDL) (Appendix A, ESI), at 100 µg/mL in water. As shown in Figure 2, the adsorption capacity of MIP_M1 towards the template was much higher than that of its analogues, exhibiting a noteworthy molecular recognition ability.

In terms of template binding efficiency, the MIPs prepared by mechanochemistry seem to perform as well or even better (>93% of the exposed template amount was bound) than the ones prepared by the conventional, solvent-based approach (in between 73 and 95% of the exposed template amount was bound). Interestingly, the chemical selectivity of the obtained imprinted polymers was more nuanced, with the best performing imprinted polymer being the one based on MAA that was obtained by mechanochemical means. Although the MIPs synthesized by the conventional solvent-based approach could discriminate PRNL and ALPRNL from the template fairly well, this was not the case with CVDL. A significant difference in terms of overall selectivity was observed when comparing the two imprinted polymers obtained by the solid-state approach, with the 2-TFMAA-based polymer having rather modest performances in terms of molecular discrimination of β-blocker analogues.

Besides the specific binding pockets resulting from the MI process, other types of “non-specific” interactions may also shape the polymers’ binding efficiency and selectivity. As it can be seen in Figure 2, a fairly strong discrimination of the β-blocker analogues by the polymers prepared under wet conditions (MIP_S1–2 and NIP_S1–2; Figure 2) was recorded. Although there were significant differences in the lipophilicity of the studied β-blocker analogues, where ATNL is the most hydrophilic and CVDL is the most lipophilic (Appendix A, ESI), a preferential partitioning in the polymeric matrix may not explain the observed chemoselectivity. However, apart from the existence of molecularly imprinted binding pockets that are sterically and functionally complementary to the employed template, the number of polar atoms defining the overall polar surface area of the β-blocker analogues might be responsible for the stronger, non-specific binding of ATNL and CVDL on both the imprinted, as well as on the non-imprinted polymer obtained under wet conditions.

This early study demonstrated the fast, efficient, and solventless synthesis of MIPs, that is readily transferable to dedicated instrumentation (planetary or vibration ball mills, etc.) that are commonly employed in mechanochemistry, offering a better control over the impact energy and higher surface areas associated with the larger milling balls. Computer-assisted design strategies commonly employed in predicting the best performing polymers in terms of binding efficiencies and selectivities might shed further light on the recorded advantages in terms of their analytical performance.

## 3. Material and Methods

### 3.1. MIP Synthesis

In the typical mechanochemical synthesis of MIP_M1 using MAA as the functional monomer, the following solid components were weighed into a 5 mL glass vial (Appendix A): MBA 374 mg, APS 46 mg, alumina 30 mg, and ATNL 26.6 mg. Next, the liquid elements were added over the solid mixture (50 μL of MAA and 50 μL of ACN), followed by two stainless steel balls (SSBs) with a diameter of 4 mm. In the case of the synthesis of MIP_M2 using 2-TFMAA as the functional monomer, the same protocol was employed, but instead of MAA, 85 mg of 2-TFMAA was weighed (Appendix A). The vial was placed on the platform of a CAPPRondo Vortex Mixer (Nordhausen, Germany) and secured with a home-made attachment system. The polymerization was achieved by vortexing at 3000 rpm for 2 h while the vial was irradiated at 365 nm using two UV lamps (18 W, L = 59 cm). Afterwards, the template was removed from the obtained polymers by washing with a MeOH/acetic acid (9:1, 10 mL) mixture three times, followed by washing trice with pure MeOH (10 mL). Finally, the polymers were dried in an oven at 40 °C for 24 h. The corresponding non-imprinted polymers were prepared following the same protocol, but in the absence of the template: NIP_M1 and NIP_M2 (Appendix A). Eventually, to ensure identical solvent exposures before the rebinding studies, the NIPs were also passed through the same washing procedure as MIP_M1/MIP_M2.

In order to evaluate the imprinting performance using the mechanochemical approach, MIPs and NIPs were also prepared under conventional wet-chemistry conditions using the corresponding aprotic, organic solvent (ACN) as the porogen (Appendix A).

Although not an ideal experimental design for a comparative study, due to the poor solubility of MBA in the porogen (ACN), in the conventional, wet MIP synthesis, the cross-linking agent MBA was replaced by EGDMA. MIPs were synthesized by weighing out 9 mg of ATNL and 9 mg of ACVA, followed by the addition of 17 μL of MAA (MIP_S1) or 29 mg of 2-TFMAA (MIP_S2), 150 μL of EGDMA, and 5 mL of ACN. After degassing the pre-polymerization mixture with nitrogen for 10 min, the photoinitiated polymerization was induced by a UV light (365 nm, 2 × 18 W, L = 59 cm) at room temperature and the reaction proceeded for 20 h. The corresponding NIPs (NIP_UV1 and NIP_UV2) were prepared in the same manner but without the template molecule (Appendix A). Template removal/polymer washing were performed in an identical manner as for the solid-state MIP/NIP counterparts.

### 3.2. Binding Experiments

To evaluate the imprinting efficiency of the obtained MIPs, 5 mg of polymer (MIP or NIP) was added to a solution containing 100 μg of a β-blocker (ATNL, PRNL, CVDL or ALPRNL) in 1 mL of ultrapure water. The mixtures were shaken in an Ika Loopster^®^ rotary shaker (IKA-Werke GmbH & Co., Staufen, Germany) at 60 rpm for 24 h. Afterwards, the suspension was centrifuged for 15 min at 10,000 rpm (Eppendorf 5340), and the supernatants were analyzed by HPLC with UV detection (Agilent 1200 series chromatographic system, Agilent Technologies, Santa Clara, United States) using an Agilent Zorbax Exclipse XDB-C18, 5 μm, 150 × 4.8 mm column. The analysis was performed at 40 °C and a flow rate of 0.8 mL/min. The mobile phase was represented by a mixture of 20 mM borate buffer, pH = 8.3, and ACN in a 20:80 (*v/v*) ratio in the case of ATNL and ALPRNL, and 60:40 (*v/v*) ratio for the analysis of PRNL and CVDL. The detection was performed at 210 nm. The adsorption capacities (*Q* (mg/g)) of the MIPs and NIPs were calculated using Equation (1).
(1)$$Q=\frac{\left(C_{0}-C_{eq}\right)V}{m}$$
where *C*_0_ (mg mL^−1^)—the initial concentration of the β-blocker in the suspension; *C_eq_* (mg mL^−1^)—the equilibrium concentration of the β-blocker in the supernatant; *V* (mL)—the volume of the suspension; and *m* (g)—the mass of the MIP/NIP.

Another measure demonstrating the successful imprinting of the template is the imprinting factor (IF), which is calculated according to Equation (2):IF = *Q_MIP_*/*Q_NIP_*(2)
where *Q_MIP_* (mg/g) represents the adsorption capacity of the MIP and *Q_NIP_* (mg/g) represents the adsorption capacity of the NIP.

### 3.3. MIP Characterization

The MIP and NIP surface morphologies were analyzed by scanning electron microscopy (SEM). Polymer particles were immobilized on aluminum stubs using carbon double-sided adhesive tabs (Electron Microscopy Sciences, Hatfield, PA, USA). The samples were coated with 10 nm of gold in a Polaron E–5100 plasma-magnetron sputter coater (Polaron Equipment Ltd., Watford, UK) in the presence of argon (45 s at 2 kV and 20 mA) prior to obtaining the SEM images. Ultrastructural images of the particles were obtained using a Hitachi SU8230 scanning electron microscope (Hitachi, Tokyo, Japan) at 30 kV and different magnification powers.

The ATR-FT-IR spectra of the pure starting materials, as well as the washed and dried reaction products were recorded using a Jasco FT-IR 4100 spectrometer (Jasco Inc., Easton, PA, USA) equipped with a ZnSe crystal for the direct acquisition of the attenuated total reflection (ATR) infrared spectra of the solid samples. The spectra were recorded in the range of 4000–550 cm^−1^ with a spectral resolution of 1 cm^−1^.

### 3.4. Comparison of DOZN 2.0 Greenness Scores for the Wet and Solid-State Batch Synthesis of Imprinted Polymers

DOZN 2.0 (Merck^®^, Darmstadt, Germany) greenness scores were calculated for the different mechanochemistry and solution-based processes for the synthesis of the imprinted and non-imprinted polymers. The synthesis procedure, raw material data based on manufacturing information (Globally Harmonized System, GHS), and corresponding hazard scores (Safety Data Sheets, SDS) were used as required inputs. An overall aggregate (greenness) score, along with individual scores against the 12 green chemistry principles arranged in 3 groups ((i) increased energy efficiency, (ii) improved resource use and (iii) reduced human and environmental hazards) were calculated for all synthesis routes.

## 4. Conclusions

Herein, we reported a solvent-free mechanochemical synthesis method for MIPs via liquid-assisted grinding. The successful synthesis of the imprinted polymer was functionally demonstrated by measuring its template rebinding capacity and the selectivity of the molecular recognition process in comparison with the polymers obtained by the conventional, non-covalent molecular imprinting process in liquid media. Even without specific optimization of the reaction mixture (molar ratios of the key participants), this early study demonstrated similar binding capacities towards the template molecules and superior chemoselectivity of these polymers compared to those obtained using the conventional MIP synthesis approach. The adoption of green chemistry principles with all their inherent advantages in the synthesis of MIPs not only alleviates potential environmental and health concerns associated with their analytical (e.g., selective adsorbents) and biomedical (e.g., drug carriers or reservoirs) applications, but it might also offer a conceptual change in molecular imprinting technology. Future studies, besides gaining a deeper understanding of the influence of different mechanochemical variables on the molecular recognition properties of the resulting imprinted polymer, will also need to answer to a series of key questions, such as what is the range of template molecules that are compatible with the mechanochemical synthesis of MIPs? What is the extent of the unwanted degradation of the employed monomers, cross-linkers, and template under the employed experimental conditions? If or to what extent the template molecule is covalently binding to the polymeric scaffold? Is mechanochemistry transferable to other MIP synthetic approaches (i.e., reversible addition-fragmentation chain transfer (RAFT), controlled radical polymerization (atom transfer radical polymerization (ATRP)), etc.?

## Figures and Tables

**Figure 1 ijms-25-05504-f001:**
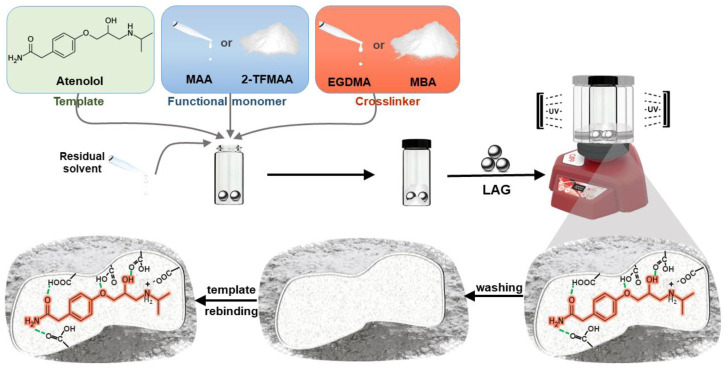
The synthesis process via liquid-assisted grinding and evaluation of the atenolol-imprinted polymers.

**Figure 2 ijms-25-05504-f002:**
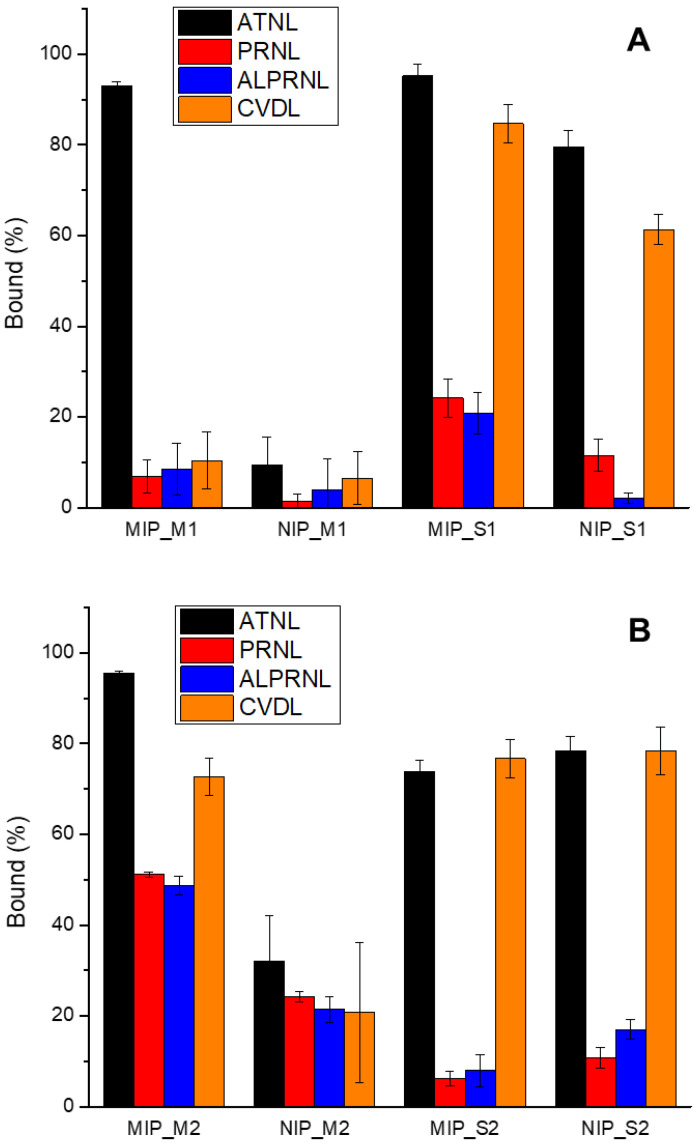
Selectivity binding studies in aqueous media using (**A**) MAA-based MIP and (**B**) 2-TFMAA-based MIP.

**Table 1 ijms-25-05504-t001:** Comparisons of greenness of the solid-state versus solution-based batch synthesis approaches for imprinted and non-imprinted polymers.

	Improved Resource Use	Reduced Human and Environmental Hazards	Aggregate
MIP_M1	1.17	0.86	1
MIP_M2	0.89	0.72	1
NIP_M1	1.68	1.15	1
NIP_M2	3.64	2.15	3
MIP_S1	9.77	17.49	12
MIP_S2	9.18	16.45	11
NIP_S1	6.73	12.29	8
NIP_S2	9.16	16.45	11

## Data Availability

Data is contained within the article and Appendix A.

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
