# Peer review of "Unlocking New Avenues: Solid-State Synthesis of Molecularly Imprinted Polymers"

_ijms, 2024, doi:10.3390/ijms25105504_

Round 1
Reviewer 1 Report
Comments and Suggestions for Authors
The paper discussing solid-state synthesis of MIPs is an interesting read and could vastly improve the 'green-ability of MIP synthesis.
Below are my comments:
1. Regarding binding results:
a.) Binding was conducted using water. It is important therefore to measure the pH before binding to ascertain the predominant form of the analyte. It is equally important to note the pKa of the analytes as this can affect the binding mechanism - is it electrostatic or H-bonding?
b.) Using mass as the binding reference may not be accurate considering that the morphologies of the MIPs and NIPs are different. If possible, use surface area.
2. Is there a way of measuring the degree of crosslinking from the FTIR data? This is important especially if the conversion is low, i.e. 5-40%. Apart from functional monomers, the rigidity of the polymers could also affect binding. If the degree of crosslinking between MIPs and NIPS are markedly different, with NIPs less crosslinked, this could erroneously give a higher IF.
3. The first part of Results and Discussion could be reviewed and amended as there seems to be parts more appropriate to be discussed in the introduction.
4. Re Table 1. I suggest removing the column with 0 values. THis does not add any information and could simply be mentioned in the text.
Comments on the Quality of English LanguageThe paper needs a thorough typo check. There are errors on fundamental words, for example, griding instead of grinding.
Author Response
The paper discussing solid-state synthesis of MIPs is an interesting read and could vastly improve the 'green-ability of MIP synthesis.
Below are my comments:
- Regarding binding results:
a.) Binding was conducted using water. It is important therefore to measure the pH before binding to ascertain the predominant form of the analyte. It is equally important to note the pKa of the analytes as this can affect the binding mechanism - is it electrostatic or H-bonding?
Thank you for your insightful comments on our manuscript. Based on our previous research (doi: 10.1021/ac504036m; 10.1016/j.electacta.2016.09.079) and due the weakly basic nature of the studied beta-adrenergic receptor antagonists confered by the secondary amine in the alkyl side chain, within the neutral to slighly acidic range of water all investigated analytes, including the template itself (atenolol, pKa = 9.43) are present in their protonated form during the rebinding process. This protonated form may engage in electrostatic interactions with the negatively charged polymer matrix. Additionally, other weaker interactions such as hydrogen bonding, van der Waals forces, etc. are contributing and most probably modulating the overall rebinding process within the MIP.
b.) Using mass as the binding reference may not be accurate considering that the morphologies of the MIPs and NIPs are different. If possible, use surface area.
Thank you for your valuable suggestion. While we understand the concern raised about the use of mass as a reference due to these morphological differences, we believe that these variations are inherent to the imprinting process and should be considered when comparing MIPs with NIPs. These differences may play a crucial role in the binding characteristics and selectivity of the imprinted polymers.
- Is there a way of measuring the degree of crosslinking from the FTIR data? This is important especially if the conversion is low, i.e. 5-40%. Apart from functional monomers, the rigidity of the polymers could also affect binding. If the degree of crosslinking between MIPs and NIPS are markedly different, with NIPs less crosslinked, this could erroneously give a higher IF.
We acknowledge the importance of quantitatively measuring the degree of crosslinking from the FTIR data, especially in the cases of low conversion percentages. While we conducted a qualitative analysis rather than a quantitative one to demonstrate the successful incorporation of monomers into the polymer matrix, we recognize the significance of assessing the degree of crosslinking for a comprehensive understanding of the polymer structure and its binding characteristics. The differences in rigidity between MIPs and NIPs could indeed influence the binding properties, particularly if the crosslinking levels vary significantly. Although our initial FTIR spectra analysis did not reveal substantial differences in band intensities or ratios between bands for MIPs and NIPs, indicating a similar polymer composition, we appreciate the importance of further investigations into the degree of crosslinking to elucidate its impact on the binding efficiency and selectivity.
- The first part of Results and Discussion could be reviewed and amended as there seems to be parts more appropriate to be discussed in the introduction.
We have carefully reviewed the first part of the Results and Discussion section and made amendments based on your suggestions. Specifically, we have reevaluated the content to ensure that information more suited for the Introduction has been appropriately transitioned to that section.
- Re Table 1. I suggest removing the column with 0 values. THis does not add any information and could simply be mentioned in the text.
We have acted on your recommendation by removing the column containing 0 values.
Furthermore, we have revised the corresponding text to incorporate the information previously displayed in the removed column.
Comments on the Quality of English Language
The paper needs a thorough typo check. There are errors on fundamental words, for example, griding instead of grinding.
We have conducted a comprehensive review of the document and have successfully rectified all typographical errors.
Reviewer 2 Report
Comments and Suggestions for Authors
Paper title “Unlocking New Avenues: Solid-State Synthesis of Molecularly Imprinted Polymers” report a solvent-free mechanochemical synthesis of MIPs via liquid-assisted griding. The successful synthesis of the imprinted polymer has been functionally demonstrated measuring its template rebinding capacity, as well as the selectivity of the molecular recognition process in comparison with the ones obtained by the conventional, non-covalent molecular imprinting process in liquid media. The adoption of green chemistry principles with all its inherent advantages in the synthesis of MIPs, not only alleviates potential environmental and health concerns associated with their analytical (e.g. selective adsorbents) and biomedical (e.g. drug carriers or reservoirs) applications but might also offer a conceptual change in the molecular imprinting technology. I really appreciate the hard work made to prepare this article. It is highly recommended to be published in International Journal of Materials Science after very minor revision.
-The language and grammar in this paper needs polish.
-references need to be correct.
-Synthesis mechanism part should be describing in different figure.
- The Benesi–Hildebrand method can be employed to understand binding efficiency means to calculate the imprinting efficiency of the obtained MIPs.
Comments on the Quality of English Language
The language and grammar in this paper needs polish.
Author Response
Paper title “Unlocking New Avenues: Solid-State Synthesis of Molecularly Imprinted Polymers” report a solvent-free mechanochemical synthesis of MIPs via liquid-assisted griding. The successful synthesis of the imprinted polymer has been functionally demonstrated measuring its template rebinding capacity, as well as the selectivity of the molecular recognition process in comparison with the ones obtained by the conventional, non-covalent molecular imprinting process in liquid media. The adoption of green chemistry principles with all its inherent advantages in the synthesis of MIPs, not only alleviates potential environmental and health concerns associated with their analytical (e.g. selective adsorbents) and biomedical (e.g. drug carriers or reservoirs) applications but might also offer a conceptual change in the molecular imprinting technology. I really appreciate the hard work made to prepare this article. It is highly recommended to be published in International Journal of Materials Science after very minor revision.
-The language and grammar in this paper needs polish.
-references need to be correct.
Thank you for your feedback on our manuscript. We have conducted a comprehensive review of the document and the text and references were reviewed and corrected.
-Synthesis mechanism part should be describing in different figure.
We are not sure if we understood this observation correctly; nevertheless, we wanted to keep figures to a minimum, and as such the conducted mechanochemical synthesis of MIPs presented in Figure 1 is merged with the potential non-covalent interactions that may be responsible for the specific rebinding of the template.
- The Benesi–Hildebrand method can be employed to understand binding efficiency means to calculate the imprinting efficiency of the obtained MIPs.
Thank you for recommending the application of the Benesi-Hildebrand method to understand binding efficiency and calculate the imprinting efficiency of the obtained MIPs. Your suggestion will definitively be considered in a further, more extended investigation of this initial proof-of-concept study, as it could provide a more thorough understanding of the binding characteristics of the MIPs fabricated by such solventless procedures.
Comments on the Quality of English Language
The language and grammar in this paper needs polish.
We have conducted a comprehensive review of the document and have successfully rectified all typographical errors and English was revised.
Reviewer 3 Report
Comments and Suggestions for Authors
It is a very good manuscript.
I have only minor critical comments. The Introduction section is too long. It is not a review of the literature. The authors should briefly formulate a problem that they will be addressing. A similar comment: the Conclusion section should briefly answer the problem outlined in the Introduction.
The values presented in Table 1 should have a determination error.
"Reduced human and environmental hazard" requires a detailed explanation. It is something very uncommon.
I recommend presenting at least one chemical reaction reflecting the described chemistry.
Comments on the Quality of English LanguageI do not see problems.
Author Response
It is a very good manuscript.
I have only minor critical comments. The Introduction section is too long. It is not a review of the literature. The authors should briefly formulate a problem that they will be addressing. A similar comment: the Conclusion section should briefly answer the problem outlined in the Introduction.
Thank you for your kind considerations related to our study and for your useful comments. Although we agree that in general shorter introductions are more common in such experimental research, in this particular and amongst the first proof-of-concept study we considered it important to more extensively describe the current context of conventional MIP synthesis and all the potential implications of the use of solvents in comparison with the perspectives of the newly proposed mechanochemical approach.
The values presented in Table 1 should have a determination error.
We acknowledge the importance of showing determination errors in a graphic, however In calculating the greenness scores presented in Table 1, no errors are typically included.
"Reduced human and environmental hazard" requires a detailed explanation. It is something very uncommon.
A text including an explanation of the hazards posed by organic solvents on humans and the environment was added.
I recommend presenting at least one chemical reaction reflecting the described chemistry.
Polymer synthesis often relies on empirical observations, practical methods, and specific experimental conditions (e.g., temperature, pressure, catalysts) that can influence the outcome. These conditions might not be easily captured in a simple reaction equation. Furthermore, polymer synthesis typically involves complex mechanisms and pathways (e.g., radical, step-growth), each with unique reaction mechanisms and with many possible intermediates, side reactions, branching and termination which add to the complexity. Accurately representing these in chemical reaction form can be complicated and impractical. Polymer synthesis involves statistical processes where monomers are added in a random or probabilistic manner. This leads to a distribution of polymer chain lengths, architectures, and molecular weights, making exact chemical descriptions challenging.
As a conclusion, the generalized and complex nature of polymer synthesis often means that simple chemical reactions do not capture all the intricacies of the process, thus at this point we would avoid including chemical reactions reflecting the described chemistry.
Comments on the Quality of English Language
I do not see problems.
Round 2
Reviewer 1 Report
Comments and Suggestions for Authors
I thank the authors for providing responses to my comments. The responses provided for comment #1 a) and b) on binding results, and for comment #2 on crosslinking are important information and can improve readers' understanding of the results presented and set the limitation of the study. I recommend incorporating these information in the relevant parts of the discussion of results prior to acceptance of manuscript for publication.
Author Response
I thank the authors for providing responses to my comments. The responses provided for comment #1 a) and b) on binding results, and for comment #2 on crosslinking are important information and can improve readers' understanding of the results presented and set the limitation of the study. I recommend incorporating these information in the relevant parts of the discussion of results prior to acceptance of manuscript for publication.
Thank you for your kind and constructive suggestions. We have incorporated in the manuscript (section 2.1, lines 237-240 and Section 2.2, lines 278-292) the information presented as response to the Reviewer in the first round of revision.